# Zinc Finger Proteins in the Human Fungal Pathogen *Cryptococcus neoformans*

**DOI:** 10.3390/ijms21041361

**Published:** 2020-02-18

**Authors:** Yuan-Hong Li, Tong-Bao Liu

**Affiliations:** 1State Key Laboratory of Silkworm Genome Biology, Southwest University, Chongqing 400715, China; 2Chongqing Key Laboratory of Microsporidia Infection and Prevention, Southwest University, Chongqing 400715, China

**Keywords:** zinc finger proteins, *Cryptococcus neoformans*, fungal development, sexual reproduction, virulence

## Abstract

Zinc is one of the essential trace elements in eukaryotes and it is a critical structural component of a large number of proteins. Zinc finger proteins (ZNFs) are zinc-finger domain-containing proteins stabilized by bound zinc ions and they form the most abundant proteins, serving extraordinarily diverse biological functions. In recent years, many ZNFs have been identified and characterized in the human fungal pathogen *Cryptococcus neoformans*, a fungal pathogen causing fatal meningitis mainly in immunocompromised individuals. It has been shown that ZNFs play important roles in the morphological development, differentiation, and virulence of *C. neoformans*. In this review, we, first, briefly introduce the ZNFs and their classification. Then, we explain the identification and classification of the ZNFs in *C. neoformans*. Next, we focus on the biological role of the ZNFs functionally characterized so far in the sexual reproduction, virulence factor production, ion homeostasis, pathogenesis, and stress resistance in *C. neoformans*. We also discuss the perspectives on future function studies of ZNFs in *C. neoformans*.

## 1. Introduction

*Cryptococcus neoformans* is a basidiomycetous yeast pathogen that causes fungal meningitis mainly in immunocompromised individuals [1]. *C. neoformans* is a ubiquitous fungal pathogen and can be found in soil, pigeon droppings, trees, fruits, and even human skin. Spores or desiccated yeast cells can be inhaled into the human lung to produce an asymptomatic infection that develops into a dormant latent infection [2]. When the host’s immunity is impaired or reduced, the dormant form of the spores or desiccated yeast can be reactivated and disseminated from the lung to the central nervous system (CNS) to cause meningoencephalitis that is fatal without proper treatment [3]. With the increase in the number of people who have HIV or AIDS or who have received organ transplants and immunosuppressive therapy [4], *C. neoformans* infects more than one million people worldwide each year, resulting in hundreds of thousands of deaths annually [5,6].

As a human fungal pathogen, *C. neoformans* has at least three major virulence factors: capsule formation, melanin production, and growth at 37 °C, which favors the infection and the pathogenesis of *C. neoformans* [7,8]. *C. neoformans* is a heterothallic basidiomycetous fungus, having two mating types, α and **a**, and can undergo the transition from yeast-form to filamentous form by mating. During mating in *C. neoformans*, haploid cells of opposite mating types fuse to form dikaryotic filaments, leading to the formation of a basidium. After completion of meiosis inside the basidium, four chains of haploid basidiospores are produced on top of the basidium [2]. Under laboratory conditions, the *C. neoformans* cells of one mating type, e.g., α cells, can also fuse and undergo haploid fruiting to form filaments and basidiospores (see Figure 1) [2,9]. Based on capsular agglutination reactions, molecular studies, and genome sequences analysis, *C. neoformans* was classified into two varieties: *C. neoformans* var. *neoformans* (serotype D) and *C. neoformans* var. *grubii* (serotype A); a former third variety (*C. neoformans* var. *gattii*, serotype B) was reclassified as a separate species *C. gattii* [10,11,12,13]. Recently, based on the analysis of genealogical concordance, coalescence-based, and species tree approaches, some researchers proposed that the current *C. neoformans* var. *neoformans* and *C. neoformans* var. *grubii* can be recognized as separate species, *C. deneoformans* and *C. neoformans*, and the current *C. gattii* complex can be reclassified into five species [14]. Others recommend using “*C. neoformans* species complex” and “*C. gattii* species complex” to recognize genetic diversity and minimize the nomenclatural instability [15].

Up to now, many signaling pathways or factors involved in sexual reproduction and fungal virulence have been characterized and extensively studied [3,16,17,18,19,20]. However, these studies revealed that the regulation mechanism of fungal development and virulence in *C. neoformans* is very complicated and still need to be further explored.

ZNFs are zinc finger domain-containing proteins and form the most abundant proteins, serving extraordinarily diverse biological functions, such as transcriptional activation and regulation, protein degradation, signal transduction, and plenty of other functions [21]. Fungal ZNFs are best characterized in the budding yeast, *Saccharomyces cerevisiae*, where there have been more than 50 ZNFs identified in this organism based on the analysis of genome sequence data [22]. Among the ZNFs, Gal4 was the first and is the best studied ZNF and activates the genes responsible for galactose catabolism in *S. cerevisiae* [22,23]. ZNFs have also found and investigated in varieties of other fungal organisms, such as the fission yeast *Schizosaccharomyces pombe* and the human yeast-like pathogens *Candida albicans* and *C. neoformans*. In this review, we aimed to describe the identification and classification of ZNFs and summarize the current state of knowledge of ZNFs in the sexual reproduction, virulence factor production, metal homeostasis, pathogenesis, and stress resistance in the human fungal pathogen *C. neoformans*.

## 2. Zinc Finger Proteins: An Overview

ZNFs are wildly present in eukaryotes and participate in various cellular processes, including signaling transduction, gene activation and regulation, and cell differentiation. ZNFs are considered to be one group of proteins for they all harbor a common zinc finger motif, yet they display variable secondary structures and enormous functional diversity. A zinc finger is usually defined as a small, functional protein, with a structural motif that is coordinated by one or more zinc ions, and typically serves as an interactor module that binds DNA, RNA, proteins, or small molecules [21]. The zinc finger motif was initially identified as a repeated zinc-binding motif in the transcription factor IIIA (TFIIIA) in *Xenopus laevis* [24] and it was found to have a protruding “finger-like” shape a few years later by resolution of its three-dimensional solution structure [25]. Since then, numerous other ZNFs have been identified and functionally analyzed in many species, including pathogenic fungi. The majority of the ZNFs identified initially bind to DNA/RNA, suggesting a role in transcriptional and/or translational processes [21]. However, further study showed that ZNFs are not only limited to binding nucleic acids, but also are involved in many other physiological functions, such as regulating zinc sensing, chromatin remodeling, lipid binding, protein-protein interactions, and protein chaperoning [21,22]. 

ZNFs can coordinate divalent zinc ions with the combination of cysteine and histidine residues. Initially, based on the structural and functional variation of the zinc finger domain, ZNFs were categorized into at least nine types: C_2_H_2_, C_2_HC, C_3_H, C_4_, C_6_, C_3_HC_4_, C_2_HC_5_, C_4_HC_3_, and C_8_, in which C and H represent cysteine and histidine, respectively [26,27,28]. These nine types of ZNFs are summarized in Table 1. Most zinc finger proteins contain only one type of zinc finger, while other evolutionarily-conserved structural features were also identified outside the zinc finger domain in some ZNFs. The C_2_H_2_-type finger, represented by TFIIIA [24], is the classic zinc finger that has been best studied and is present in many transcription factors and other DNA-binding proteins [21]. C_2_HC-type zinc finger domains, also referred to as retroviral-type (RT) zinc finger sequences, are required for viral genome packaging and RNA or single-stranded DNA binding in eukaryotes [29,30,31,32,33]. The C3H family proteins are divided into 18 groups based on the different amino acid spacing numbers between C and H in zinc finger motif [26,34,35]. The C4 family includes seven types: GATA, FYVE, Tim10/DDP, LSD1, A20, TFIIB, and Zn-finger in Ran-binding protein [36]. GATA-1 was the first member of C4 family proteins characterized by the recognition of the GATA DNA sequence [37]. C_3_HC_4_-type finger, also termed RING finger, can be categorized into seven types with different conserved motifs, such as RING-H2, RING-HC, RING-v, RING-D, RING-S/T, RING-G and RING-C2 [38,39,40]. Many proteins with a RING finger play a vital role in the ubiquitination pathway [41,42]. The C_2_HC_5_ motif, also referred to as LIM, was first identified in the cell lineage gene lin-11 in *Caenorhabditis elegans* [43].

Then, the ZNFs were categorized into eight-fold groups based on the structural properties in the folded domain by a more systematic method [49]. Among these fold groups, the first three are Cys2His2-like, treble clef, and zinc ribbon, comprising the majority of zinc fingers [49,50]. Recently, based on the zinc-finger domain structure, 30 types of ZNFs were classified and were approved by the Human Genome Organisation (HUGO) Gene Nomenclature Committee [51].

## 3. ZNFs Identified in *C. neoformans*

With the increase in the immunocompromised population resulting from AIDS and widespread immunosuppressive therapy, *C. neoformans* has become the main fungal pathogen in individuals with impaired immunity [1,52]. To better elucidate the genome basis for virulence in *C. neoformans*, the *Cryptococcus* research community has sequenced the genomes and/or transcriptomes of the two varieties of *C. neoformans*: *C. neoformans* var. *neoformans* (JEC21, serotype D) and *C. neoformans* var. *grubii* (H99, serotype A) [52,53], providing the data for the comparative analysis of the genomes of the isolates between the two serotypes. The *Cryptococcus* genome contains a large number of genes annotated to encode ZNFs. When we performed a keyword search combined with “zinc finger” or “zinc-finger”, 127 and 129 genes encoding ZNFs were identified and characterized, according to the combination of cysteines and histidines, which coordinated Zn^+2^ ions in *C. neoformans* var. *neoformans* and *C. neoformans* var. *grubii* genome database, respectively (See Appendix A). Based on the distinct structural properties of the zinc finger motif, we have categorized the ZNFs identified in the *C. neoformans* genome database of both varieties (see Table 2). The functional annotation of ZNFs shows their critical biological roles in a diverse range of cellular processes, such as transcriptional regulation, protein transportation and degradation, chromatin remodeling, mRNA processing, and DNA binding and repairing. Thus, ZNFs may play crucial roles in fungal growth and development in *C. neoformans*.

## 4. ZNFs Regulate Sexual Reproduction in *C. neoformans*

ZNFs form one of the largest families of transcriptional regulators, playing an essential role in the growth and development in eukaryotes. Over the past two decades, multiple signaling pathways, such as pheromone-sensing, light-sensing, cAMP/protein kinase A(PKA), calcineurin, and high-osmolarity glycerol (HOG) signaling pathways, regulating sexual reproduction and hyphal morphogenesis in *C. neoformans*, were identified and characterized via genetic and genomic approaches [18,54,55,56,57]. 

### 4.1. ZNFs Regulate Cell Fusion during Mating in *C. neoformans*

As a heterothallic basidiomycete, *C. neoformans* can undergo a dimorphic transition to a filamentous growth form by mating and monokaryotic fruiting (Figure 1). Cell fusion is the first step of α-**a** bisexual mating or α-α unisexual mating and can be regulated by the pheromone pathway. In *C. neoformans*, the Cpk1 (*C. neoformans* protein kinase 1) MAPK (mitogen-activated protein kinase) signal transduction cascade is a highly-conserved signaling circuit controlling the dimorphism transition during bisexual and unisexual reproduction [54]. However, the downstream targets of the pathway are mostly unknown. Three ZNFs, encoding potential transcription factors, were identified in the process of identification of downstream targets of the Cpk1 MAPK pathway: *ZNF1*α, located in the mating type locus, Znf2, and Znf3, residing in another genomic region [58,59,60]. Functional analysis showed that Znf1 is not crucial for *Cryptococcus* dimorphic hyphal growth because Znf1 mutations in both mating types in the var. *neoformans* strain backgrounds did not suppress cell fusion or hyphal formation during bisexual and unisexual mating [58]. Protein sequence analysis showed that Znf2 contains four C_2_H_2_ zinc finger domains, indicating that it may be a transcription factor. Loss of Znf2 locks cells in the yeast phase, while overexpression of this regulator drives hyphal growth, indicating that Znf2 is critical for filamentation and hyphal morphogenesis after the cell-cell fusion event during mating [58,61,62,63]. Lin et al. further proved that Znf2 regulates cryptococcal cellular differentiation in coordination with the chromatin remodeling SWItch/sucrose non-fermentable (SWI/SNF) complex [64]. Znf3 contains three C_2_H_2_ zinc finger domains and sexual reproduction analysis showed that Znf3 plays an essential role in both unisexual and bisexual reproduction [59] (Figure 1). Another study showed that Znf3 promotes cell fusion and pheromone production, not in the pheromone signaling cascade, but in a parallel and independent pathway [59]. Surprisingly, Znf3 was also found to be necessary for mitotic- or sex-induced RNAi silencing in *Cryptococcus* pathogenic species [59,65].

### 4.2. ZNFs Regulate Mating Hyphal Growth in *C. neoformans*

After fusion of the cells of the opposite mating types during *C. neoformans* mating, the resulting dikaryotic cell initiates filamentous growth, with two parental nuclei migrating coordinately in the hyphae. During monokaryotic fruiting, two cells of the same mating type, e.g., α cells, fuse to become diploid α/α cells to form the diploid monokaryotic hyphae and initiate filamentous growth. 

The calcineurin signaling pathway and light-sensing pathway are involved in the regulation of hyphae growth in *C. neoformans*. Calcineurin is a highly-conserved Ca^2+^/calmodulin-dependent protein phosphatase and involves stress responses, fungal morphogenesis, and virulence in the three major human fungal pathogens [66,67]. Although it is not essential for initial cell fusion, the calcineurin signaling pathway is vital for hyphal elongation during both the bisexual and unisexual reproduction of *C. neoformans* [55]. Crz1(the calcineurin-responsive zinc finger transcription factor 1) is a downstream target of the calcineurin pathway and orchestrates distinct cellular processes in *C. neoformans*. Fu et al. demonstrated that Crz1 is essential for the sporulation process although it is not required for yeast-hyphal morphological transition during unisexual reproduction of *C. neoformans* [68]. Furthermore, Jung et al. showed that Crz1 plays roles in fungal virulence and stress survival in *C. neoformans* [69].

Light inhibits mating and monokaryotic fruiting of *C. neoformans*, but the molecular mechanism involved remained unclear. Research from two different groups has shown that filamentation associated with mating and monokaryotic fruiting is explicitly inhibited by blue light [70,71]. *C. neoformans* orthologues of *Neurospora crassa* white collar genes (*wc-1* and *wc-2*) in light sensing were identified and named independently as *CWC1*/*BWC1* (*Cryptococcus WC-1 homologue*/*Basidiomycete White Collar 1*) and *CWC2*/*BWC2* (*Cryptococcus WC-2 homologue*/*Basidiomycete White Collar 2*) [70,71], and a conserved GATA-type zinc finger DNA-binding domain was found in Cwc2/Bwc2 [70]. Functional analysis showed that both *BWC1* and *BWC2* are essential for mediating light inhibition of mating and haploid fruiting [70,71] and the overexpression of these genes further inhibits filamentation upon light treatment in *C. neoformans* [70]. Yeh et al. further confirmed that the GATA-type zinc finger DNA-binding domain is crucial for the function of the Cwc2 protein by generating partially-deleted versions of the Cwc2 [72]. 

### 4.3. ZNFs Regulate Basidiospore Formation during Mating or Monokaryotic Fruiting in *C. neoformans*

In the late stage of *C. neoformans* mating or monokaryotic fruiting, a basidium formed at the tip of the mating hyphae and four chains of basidiospores are produced on top of the basidium. Signaling pathways, such as the MAPK pathway, ubiquitin-proteasome pathway, and quorum-sensing pathway, are involved in the regulation of basidiospore formation in *C. neoformans*. 

The MAPK cascade, also referred to as the pheromone response pathway, regulates the dimorphic switch in *S. cerevisiae*, providing a framework for studying morphogenesis in a variety of fungal species. *STE12* (Sterile gene 12) encodes a downstream C_2_H_2_-type transcription factor of the pheromone response MAPK cascade, controlling fungal development in *S. cerevisiae* [73]. *STE12* homologs, *STE12α* and *STE12a*, have also been identified in *C. neoformans* and are encoded by the mating-type locus [74,75,76]. Although disruption of *STE12α* or *STE12a* does not abolish pheromone sensing or mating, deletion of the gene results in defective monokaryotic fruiting in *C. neoformans* [74,75,77]. 

F-box proteins are the key recognition subunit of the E3 ligase complexes that take part in protein ubiquitination and degradation [78]. The F-box protein (Fbp1) in *C. neoformans* is essential for fungal sexual reproduction as basidiospore production was blocked in bilateral mating between *fbp1*Δ mutants, even though normal dikaryotic hyphae were observed during mating [19]. One C_2_H_2_-type zinc finger protein (Zfp1) was identified as a potential substrate of Fbp1 by IP pulldown and LC-MS/MS methods. Functional analysis showed that Zfp1 is also essential for fungal sexual reproduction as deletion or overexpression of Zfp1 abolishes meiotic sporulation of *C. neoformans*, which indicates Zfp1 is essential for regulating meiosis during mating [79]. 

Bacterial quorum sensing is a well-characterized cell-cell communication system to coordinate population density-dependent changes in behavior [80,81]. Recently, Tian et al. identified a fungal quorum-sensing peptide (Qsp1) and showed that it serves as an essential signaling molecule for both forms of sexual reproduction in *C. neoformans* [82]. The authors further identified the C_2_H_2_-type zinc finger regulator (Cqs2) as a critical component of the Qsp1 signaling cascade during both bisexual and unisexual reproduction. Loss of Cqs2 significantly impaired bisexual and unisexual filamentation and abolished meiotic sporulation during both bisexual and unisexual reproduction, indicating ZNFs may play a role in bridging sexual production and the perception of surrounding stresses [82]. 

## 5. ZNFs Regulate the Virulence in *C. neoformans*

*C. neoformans* can infect the human CNS, causing meningitis and killing hundreds of thousands of people each year. Since it is a significant yeast pathogen with genetic tractability, extensive studies have been conducted on the pathogenesis in *C. neoformans*. Virulence factors and signaling pathways that are crucial for fungal virulence have been identified and extensively studied in *C. neoformans* [16,17,83,84,85]. ZNFs has also been shown to play an essential role in fungal virulence in *C. neoformans* [76,77,86,87,88,89].

### 5.1. ZNFs Regulate Virulence Factors Production in *C. neoformans*

As a human fungal pathogen, *C. neoformans* has three classical virulence factors: melanin production, capsule formation and growth at 37 °C, which favors the infection and the pathogenesis of *C. neoformans*. *Cryptococcus STE12α* encodes a protein containing both homeodomain and zinc finger regions in *MATα* cells of *C. neoformans* [86]. Virulence test showed that the virulence of a *ste12α* mutant was significantly reduced in a mouse model as deletion of *STE12α* resulted in a reduction in the capsular size of yeast cells in *C. neoformans* [76,77]. Mutation in the homeodomain of *STE12α* reduces DNA binding ability, mating frequency, and haploid fruiting capability but increases virulence and capsule size of yeast cell in *C. neoformans*. In contrast, mutation in the zinc fingers region also resulted in virulence attenuation, capsule size reduction, and decreased gene expression of capsule associated genes [86]. The second *STE12* homolog, *STE12a*, was also identified in *MATa* cells in *C. neoformans*. Deletion of *STE12a* markedly reduced virulence in mice, as is the case with *STE12α* [76]. Northern blot analysis showed that *STE12ap* regulates mRNA levels of several genes that are important for virulence, including *C. neoformans* laccase gene (*CNLAC1*) and capsule (*CAP*) genes. These results clearly showed that mating-type specific genes are not only crucial for sexual reproduction but also play an essential role in virulence of *Cryptococcus* in host tissue.

Protein kinase C-1 (Pkc1) signaling pathways is the primary cellular pathway mediating the cellular responses to environmental stresses. In *C. neoformans*, mutation of Pkc1 results in defects of cell wall integrity, aberrant capsule, decreased melanogenesis, and inability to grow at 37 °C [90]. Overexpression of the Pkc1-dependent specificity protein-1 (sp1) encoding gene *SP1* in a *pkc1*Δ mutant rescued the phenotype defects involved in virulence, including cell wall integrity, nitrosative stress, and extracellular capsule production. Besides, deletion of sp1 showed similar phenotype as *pkc1*Δ mutant, suggesting that the C-terminal ZNFs may shift from calcineurin-dependent to Pkc1-dependent transcription factors in the process of fungal evolution [87]. 

### 5.2. ZNFs Mediate Cryptococcus Crossing the Blood-Brain Barrier

*C. neoformans* can invade the CNS to cause cryptococcal meningoencephalitis, most commonly in individuals with impaired immunity. However, the molecular mechanism used by *C. neoformans* to invade the CNS is not very clear. Vu et al. identified a novel secreted metalloprotease (Mpr1) and demonstrated that it is required for *C. neoformans* to establish fungal disease in the CNS [88]. Mpr1 belongs to the M36 class of fungalysins unique to fungi, which are generally synthesized as propeptides with relatively long prodomains and highly-conserved regions within their catalytic core. Functional analysis revealed that Mpr1 promotes *Cryptococcus* to cross the blood-brain barrier (BBB) by facilitating attachment of cryptococci to the endothelium surface. Strikingly, the sole expression of cryptococcal *MPR1* in *S. cerevisiae* endowed the yeast cells with the ability to cross the BBB, which demonstrated that Mpr1 is specific in breaching the BBB and suggested that Mpr1 may function independently of the hyaluronic acid-CD44 pathway in breaching the brain endothelium [88]. Further structure-function analysis of Mpr1 showed that both mutations in the cleavage sites of the prodomain and amino acid substitutions in the HExxH motif resulted in the failure of cryptococci to cross the BBB [91]. The proteolytic activity assay showed that the implementation of Mpr1′s complete function was dependent on the coordination of zinc with the HExxH motif in the active site of Mpr1 [91].

### 5.3. ZNFs Mediate Metal Homeostasis in *C. neoformans*

Metals are essential for life and play a central role in the struggle between microbial pathogens and their hosts [92]. The uptake and utilization of the essential metals, such as iron and zinc, are strictly regulated in fungal pathogens. Iron is a critical cue for *C. neoformans* in regulating the elaboration of the polysaccharide capsule during infection. Iron-responsive transcription factor (Cir1) is a GATA-type zinc finger protein regulating iron uptake, iron homeostasis, and virulence factor expression in *C. neoformans*. Mutation of Cir1 results in the loss of capsule production, decreased cell wall integrity and membrane functions, marked growth defect at host body temperature, increased activity of laccase, and complete loss of virulence in both serotype A and D strains in *C. neoformans* [89]. Further study showed that the abundance of Cir1 is influenced by iron availability [93]. Thus, *Cryptococcus* can perceive iron as part of the disease process through the zinc finger protein Cir1, which may provide opportunities for antifungal treatment.

The regulation of zinc homeostasis is vital for fungal survival and virulence [94]. To investigate how zinc metabolism affects fungal virulence, the C_2_H_2_-type zinc finger protein Zap1, an ortholog of ZafA and Zap1p in *A. fumigatus* and *S. cerevisiae* [94,95], was identified and functionally analyzed in *C. gattii*. Disruption of Zap1 not only impaired *C. gattii* growth in zinc-limiting conditions but also decreased its virulence in a murine model of pulmonary infection, indicating ZNFs is required for proper zinc metabolism and plays an essential role in cryptococcal virulence [96].

## 6. ZNFs Regulate Stress Resistance in *C. neoformans*

*C. neoformans* is confronted with a variety of host-specific stresses, such as nutrient limitation, temperature shift, changes in oxygen/carbon dioxide levels, oxidative/nitrosative stress, and osmotic stress, upon its infection of the host lung. Signaling pathways or factors involved in the stress response have been identified and extensively studied [56,97,98]. ZNFs are also involved in the regulation of the stress response in *C. neoformans*. 

As a calcineurin-responsive zinc finger transcription factor, Crz1 has been proved to govern unisexual reproduction in self-filamentous *C. neoformans* serotype D strain. In contrast, Crz1 is also essential for cell wall damage repair, biofilm formation, and susceptibility to fluconazole in *C. neoformans* serotype A strain [99]. Moreover, the Crz1 protein was further proved to be involved in the slowdown of proliferation and survival under reduced aeration [99].

In *S. cerevisiae*, the C_2_H_2_-type ZNF Mig1 (multicopy inhibitor of GAL gene expression) is a carbon catabolite repressor responsible for the repression of genes involved in alternative carbon metabolism, respiration, and gluconeogenesis [100,101]. In contrast, the Mig1 homologue in *C. neoformans* is involved in the regulation of mitochondrial functions, such as respiration, tolerance for reactive oxygen species (ROS), and expression of genes involved in iron consumption and acquisition [102]. Remarkably, *MIG1* deletion increased fluconazole susceptibility in *C. neoformans*, highlighting an association between drug susceptibility and mitochondrial dysfunction and providing possible new targets for antifungal drug development [102].

The zinc finger protein Zfp1 is a potential substrate of the F-box protein Fbp1 and is involved in fungal sexual reproduction and virulence in *C. neoformans* [79]. Stress response assays showed that deletion or overexpression of Zfp1 increased susceptibility to sodium dodecyl sulfate (SDS), but not Congo red, demonstrating that Zfp1 may govern cell membrane integrity, which may be one of the reasons the virulence declined in a murine systemic-infection model [79].

mRNA reprogramming is necessary for successful stress adaptation in *C. neoformans*. Two small zinc knuckle RNA binding proteins, gluconeogenic growth suppressor 2 (Gis2) and zinc-finger protein 9 (Znf9), were identified by mass spectrometry in *C. neoformans*. Loss of Gis2 or Znf9 not only increased susceptibility to cobalt chloride, fluconazole, and oxidative stress but also repressed the levels of sterols in *gis2*Δ or *gis2*Δ *znf9*Δ double mutants although transcriptional induction of C-4 sterol methyl oxidase gene (*ERG25*) was similar to that of the wild type [103]. These results clearly showed that the zinc finger proteins Gis2 and Znf9 are necessary for stress resistance and ergosterol biosynthesis in *C. neoformans*.

The ZNFs described in the literature and discussed in this review are summarized in Table 3.

## 7. Conclusions and Future Directions

ZNFs are one of the most abundant groups of proteins and constitute the largest family of transcription factors that play important roles in prokaryotes and eukaryotes. Recent studies have shown that ZNFs play crucial roles in fungal development and differentiation, ion homeostasis, cell wall integrity, drug susceptibility, and virulence in the human fungal pathogen *C. neoformans*. In this review, the various roles of ZNFs in *C*. *neoformans* were summarized.

Although many ZNFs have been studied and have been shown to play important roles in fungal development, stress response, and virulence in *C. neoformans*, there are still some areas that should be studied: first, the molecular mechanisms that link ZNFs to their biological roles; second, only 20 ZNFs have been identified and functionally analyzed in *C. neoformans*; third, most of the ZNFs that have been functionally characterized in *C. neoformans* belong to the C_2_H_2_ family; and fourth, although many studies have investigated the role of ZNFs in resistance to antifungal drugs, new antifungal drug development is still an issue. Thus, it is also urgent to investigate the molecular basis of ZNFs in the interaction between *C. neoformans* and host cells since most ZNFs are involved in fungal virulence in *C. neoformans*. At last, it will be of great interest in the future to see whether some ZNFs could be a target to develop new antifungal drugs.

## Figures and Tables

**Figure 1 ijms-21-01361-f001:**
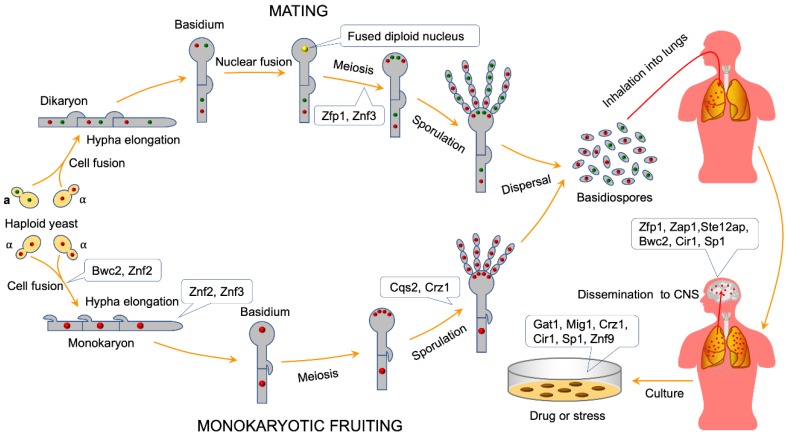
The biological function of zinc finger proteins (ZNFs) in *C. neoformans*. Under induction of nutrient-limiting conditions, α and **a** cryptococcal yeast cells can fuse and form dikaryotic filaments, leading to the formation of a basidium in which meiosis occurs to produce four chains of haploid basidiospores. Under laboratory conditions, *C. neoformans* cells of the α mating type can also fuse and undergo monokaryotic fruiting to produce filaments and basidiospores. The basidiospores can be inhaled into the lungs to establish a dormant infection or disseminate to the central nervous system to cause meningitis in humans. ZNFs are involved in different fungal development stages or processes, such as cell fusion, filamentation, sporulation, virulence, stresses, and light responses, in *C. neoformans*.

**Table 1 ijms-21-01361-t001:** The nine types of zinc finger domains.

Zinc-Binding Domain Type	Consensus Sequence	Function	Examples	References
C_2_H_2_	C-X_2-4_-C-X_12_-H-X_3-5_-H	Transcription, nucleic acid binding	TFIIIA	[24]
C_2_HC	C-X_2_-C-X_4_-H-X_4_-C	Genome packaging, single-stranded nucleic binding	Retroviral nucleocapsid	[29,30,31,32,33]
C_3_H	C-X_6-14_-C-X_4-5_-C-X_3_-H	RNA binding	Nup475	[26,34,35]
C_4_	C-X_2_-C-X_17_-C-X_2_-C	DNA binding	GATA-1	[36]
C_6_	C-X_2_-C-X_6_-C-X_6_-C-X_2_-C-X_6_-C	DNA binding	GAL4	[44]
C_3_HC_4_	C-X_2_-C-X_9-39_-C-X_1-3_-H-X_2-3_-C-X_2_-C-X_4-48_-C-X_2_-C	Protein-protein interaction, nucleic acid binding	RING finger	[38,39,40,45]
C_2_HC_5_	C-X_2_-C-X_17-19_-H-X_2_-C-X_2_-C-X_2_-C-X_16-20_-C-X_2-3_-C-C-X_2_-C-X_17_-C-X_2_-C	Protein-protein interaction, DNA binding	lin-11, isl-1 and mec-3	[43]
C_4_HC_3_	C-X_2_-C-X_11-21_-C-X_2_-C-X_4_-H-X_2_-C-X_14-17_-C-X_2_-C	Transcription, DNA binding	Requiem, ALL-1	[46,47]
C_8_	C-X_2_-C-X_13_-C-X_2_-C-X_15_-C-X_5_-C-X_12_-C-X_4_-C	Oligomerization, DNA binding	steroid-thyroid receptor	[48]

C: cysteine; H: histidine; X: any amino acid.

**Table 2 ijms-21-01361-t002:** Classification of ZNFs identified in genomes of *C. neoformans* var. *neoformans* JEC21 and *C. neoformans* var. *grubii* H99.

ZNF Type	No. of ZNFs in JEC21	%	No. of ZNFs in H99	%
C_2_H_2_	28	22.0	28	21.7
C_4_	28	22.0	32	24.8
C_3_HC_4_	21	16.5	23	17.8
C_3_H	13	10.2	12	9.3
C_2_HC	10	7.9	12	9.3
C_6_	5	3.9	4	3.1
C_4_HC_3_	2	1.6	1	0.8
C_2_HC_5_	1	0.8	1	0.8
Combination	6	4.7	6	4.7
Others	13	10.2	10	7.8
Total	127	100	129	100

**Table 3 ijms-21-01361-t003:** Summary of zinc finger proteins described in the literature and discussed in this review.

Name	Gene ID	Zinc Finger Type	Description and Function	References
Bwc2	CNE01220	ZnF_C4	Sexual filamentation and fungal virulence	[70,71]
Cir1	CNAG_04864CNJ02920	ZnF_C_4_	Iron uptake, iron homeostasis, and virulence	[89,93]
Cqs2	CNF00370	Znf_C_2_H_2_	Important component of the Qsp1-signaling cascade required for sexual reproduction	[82]
Crz1	CNA01450	Znf_C_2_H_2_	Regulates the sporulation process	[68]
Crz1	CNAG_00156	Znf_C_2_H_2_	Cell integrity; Proliferation and survival under reduced aeration; Biofilm formation and susceptibility to fluconazole.	[99]
Gat1	CNAG_00193	ZnF_C_4_	Nitrogen uptake and metabolism	[104]
Gis1	CNAG_02338	Znf-C_2_HC	Sterol biosynthesis and fluconazole and oxidative stress sensitivity	[103]
Mig1	CNAG_06327	Znf_C_2_H_2_	Mitochondrial function and azole drug susceptibility	[102]
sp1	CNAG_00156	Znf_C_2_H_2_	Cell wall stability, nitrosative stress, extracellular capsule production, and fungal virulence	[87]
*STE12αp*	CND05810	Znf_C_2_H_2_	Haploid filamentation and fungal virulence	[74,75,77]
*STE12ap*		Znf_C_2_H_2_	Capsule formation and fungal virulence; Capsule-associated genes expression;	[76,86]
Zap1	CNBG_4460	Znf_C_2_H_2_	Growth in zinc-limiting conditions Expression of ZRT, IRT-like protein (ZIP) zinc transporters and distinct zinc-binding proteins; Fungal virulence	[96]
Zfp1	CNAG_07329	Znf_C_2_H_2_	Sexual reproduction and fungal virulence	[79]
Znf1	CND05720	Znf_C_4_HC_3_	Not essential for *Cryptococcus* dimorphic hyphal growth	[58]
Znf2	CNG02160	Znf_C_2_H_2_	Master activator of the yeast to hyphal transition; Key regulator for hyphal growth	[58,61,62,63]
Znf3	CNK01880	Znf_C_2_H_2_	Hyphal development during unisexual and bisexual reproduction	[59]
Znf3	CNAG_02700	Znf_C_2_H_2_	Link between RNAi silencing and sexual development	[65]
Znf9	CNAG_01273	Znf_C_2_HC	Sterol biosynthesis; Fluconazole and oxidative stress sensitivity	[103]

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
