# Peer review of "Zinc Finger Proteins in the Human Fungal Pathogen Cryptococcus neoformans"

_ijms, 2020, doi:10.3390/ijms21041361_

Round 1
Reviewer 1 Report
Li and Liu have reviewed the various functions of zinc finger (ZNF) proteins and its role in the fungal pathogen Cryptococcus neoformans. This review updates the current knowledge of ZNF, in particular describes the variety of ZNF currently been characterized in C. neoformans, as well as describe the most recent roles of ZNF in virulence, mating, and stress tolerance for this particular yeast. The authors must address the two major questions before manuscript can be accepted.
One major question that should be addressed is the homology of characterized ZNF in C. neoformans to host (murine or human). The discussion of how ZNF could be a potential target of for anti-fungal targets and host-pathogen interactions was raised in the conclusions and future directions section. Various sections of the manuscript pointed out that ZNF is highly conserved in eukaryotic organisms. The authors should address whether ZNF could be a feasible target for anti-fungal treatments.
The other question is: Of the known 20 ZFN is their role conserved in other fungal pathogens? Alternatively, are there any orthologs that could be cited so the manuscript can be broadened to other fungal organism- this could increase readership by other medical mycologist.
There are run-on sentences (Line 185-188) and repeating sentences that require attention.
Citing more current manuscripts describing the most recent nomenclatures for Cryptococcus species are needed on a recent review to reflect the current literature (fungal genetics and biology, May 2015, Hagen et al.)
There are similar sentences (ie. line 30-33 and 230-232) that may need to be we-written to avoid redundancies.
The use of recently may want to be reconsidered, example: line 175: Recently, researchers ... discussed the inhibition of filamentation to blue light doesn't have a citation. If it discusses the next citations 62-63 they are from 2005.
other minor concerns:
Line 17 - did you imply fatal instead of fetal?
Line 177 - white collar doesn't need to be italicized
Line 219-220, cause and causing- could use other word ex: resulting
Line 332-333 should be prokaryotes and eukaryotes
reference 58, 75 - Cryptococcus needs to be italicized
reference 65 - Saccharomyces cerevisiae
Table 2 line 210- not sure if the symbol was changed
May I suggest to have section 5.1 virulence factors then 5.3 BBB prior to 5.2 metal homeostasis to improve the flow?
Reviewer 2 Report
Zinc finger proteins in the human fungal pathogen Cryptococcus neoformans
This manuscript is an interesting review; however, authors should improve it before publication. The review is an updated description of the knowledge about zinc finger proteins (ZNFs) in the Cryptococcus neoformans that affects the human being, mainly immunosuppressed and how some mutations in the ZNFs could modify the immune response and be the key to developing new therapies.
Important issues
The title is clear and concise. Although the manuscript has an adequate structure, the main objective of the summary does not match the objective of the study or the information described. The introduction is adequate however it is observed that the same ZNF information is repeated over several times, this information and the effect of the Cryptococcus neoformans on human beings should be explored further. The classification on the ZNF could be more visible / objective in a table. With respect to abbreviations and their definitions they should be homogenized, as some appear without their definition and others appear first in capital letters and then in lower case. Between words and [] there must be a space. Figure 1 is not found. Tables 1 and 2 should be improved. The conclusion should refer only to the importance of this review is not a summary of the article. The suggestions are adequate, although they should be improved.
The bibliography is current, however it could be improved. For example, the review of this article is recommended: G3 (Bethesda). 2018 Feb 2; 8 (2): 643-652. doi: 10.1534 / g3.117.300444. Had1 Is Required for Cell Wall Integrity and Fungal Virulence in Cryptococcus neoformans.
Minor issues
The manuscript is clear and easy to read, but it can be improved. This manuscript must be written correctly. They should improve the use of their references; It is too concise and lacks basic information to understand the importance of this study. The conclusions are not suitable for its objectives.
Round 2
Reviewer 1 Report
I would like to thank the authors for addressing concerns raised, however I do believe the manuscript still requires attention prior to publication, such as writing style must be addressed.
in the abstract the use of serotypes A and B in line 18 gave the impression that classification and identification of ZNF would be compared between the two. I would omit this and just discuss C. neoformans.
line 7 - extensively existed is an unusual way to state sentence, please rephrase statement
line 10 - readers may not understand the implication of dormant form ... need to add of spores/desiccated yeast
line 19- use of growth is redundant in both yeast-form and filamentous growth form
During the description of sexual reproduction please refer to figure 1.
line 72- refrain from using the term 'so on'.
line 105- please cite paper that discusses the 8 fold form of ZNFs
line 154 - please replace further with another
line 176- spell out N. crassa
the authors still discuss 'recent' manuscripts from 2014 - citation 90 for instance, this is also in line 220, but no citation was listed
line 228-231 needs rephrasing
line 236 CNLAC1 then line 242-244 gene Cn SP1 is written in a different format, please maintain consistency
I like the addition of Table 3, however there's no mention of Bwc1, Bwc2 and it was listed in Figure 1.
I would suggest to add description of yellow dot (nuclear fusion) on Figure 1.
Reviewer 2 Report
As a review of a specific topic, although it could be further enhanced, the manuscript has been rewritten and improved. However, it needs some small changes to be published. These changes appear in the attached pdf.
